# Nationwide and long-term epidemiological research of snakebite envenomation in Taiwan during 2002–2014 based on the use of snake antivenoms: A study utilizing National Health Insurance Database

Jen-Yu Hsu[1,2,3], Shu-O Chiang[4], Chen-Chang Yang[1,5], Tan-Wen Hsieh[6], Chi-Jung Chung[7,8], Yan-Chiao Mao [9,10,11,12] *

1 Department of Occupational Medicine and Clinical Toxicology, Taipei Veterans General Hospital, Taipei, Taiwan, 2 Office of Preventive Medicine, Centers for Disease Control, Ministry of Health and Welfare, Taipei, Taiwan, 3 School of Medicine, College of Medicine, National Yang Ming Chiao Tung University, Taipei, Taiwan, 4 ESTAT Statistical Consulting Co., Ltd., Taipei, Taiwan, 5 Institute of Environmental and Occupational Health Sciences, College of Medicine, National Yang Ming Chiao Tung University, Taipei, Taiwan, 6 Institute of Public Health, College of Medicine, National Yang Ming Chiao Tung University, Taipei, Taiwan, 7 Department of Public Health, College of Public Health, China Medical University, Taichung, Taiwan, 8 Department of Medical Research, China Medical University Hospital, Taichung, Taiwan, 9 PhD Program in Medical Biotechnology, College of Medical Science and Technology, Taipei Medical University, Taipei, Taiwan, 10 School of Medicine, National Defense Medical Center, Taipei, Taiwan, 11 Department of Post-Baccalaureate Medicine, College of Medicine, National Chung Hsing University, Taichung, Taiwan, 12 Division of Clinical Toxicology, Department of Emergency Medicine, Taichung Veterans General Hospital, Taichung, Taiwan

* doc1385e@gmail.com

## Abstract

### Introduction

In Taiwan, six venomous snake species with medical importance have been found; however, long-term epidemiological data of snakebite envenomation (SBE) is lacking. This study aimed to explore the epidemiology of SBE based on the distribution and use of different antivenoms in different parts of Taiwan to facilitate the development of prevention strategies and resource allocation.

### Methods and results

This retrospective study was conducted using the Taiwan National Health Insurance Research Database from 2002 to 2014. A total of 12,542 patients were treated with antivenoms. The directly standardized cumulative incidence was 3.6 cases per 100,000 individuals based on the 2000 World Standard Population. The incidence of SBEs peaked in the summer (35.9%). The relative risk (RR) of male patients versus female patients was 2.5 ($p <$ 0.0001). The RRs of patients aged 18–64 and ≥65 years versus those aged <18 years were 6.0 ($p <$ 0.0001) and 14.3 ($p <$ 0.0001), respectively. Furthermore, the RR of eastern Taiwan versus northern Taiwan was 6.8 ($p <$ 0.0001). The RR of agricultural workers versus

**Data Availability Statement:** All relevant data are within the manuscript and its Supporting information files.

**Funding:** This study was partly supported by National Science and Technology Council (MOST 109-2320-B-075A-001) and Taichung Veterans General Hospital Research Fundings (TCVGH-1067205C; TCVGH-1127201B). Y.C.M. received the fundings. The funders had no role in study design, data collection and analysis, decision to publish, or preparation of the manuscript.

**Competing interests:** The authors have declared that no competing interests exist.

laborers was 5.5 ($p < 0.0001$). Compared with patients envenomed by *Trimeresurus stejnegeri stejnegeri* or *Protobothrops mucrosquamatus*, those envenomed by *Naja atra* or *Bungarus multicinctus multicinctus* were more likely to occur in central (adjusted odds ratio [aOR] = 2.6, $p < 0.0001$) or southern (aOR = 3.2, $p < 0.0001$) Taiwan, but less frequently among agricultural workers (aOR = 0.6, $p < 0.0001$). The overall case-fatality rate was 0.11%.

## Conclusions

Among Asian countries, Taiwan had low incidence and case-fatality rates of SBE. Risk factors included male gender, old age, summer season, being in eastern Taiwan, and being an agricultural worker. Differences of the epidemiological findings between snake species should be focused on when developing strategies for snakebite prevention.

### Author summary

SBE is a neglected tropical disease of highest priority to be aware of, and its epidemiology varies widely worldwide. This study explored the nationwide and long-term epidemiology of SBE based on the use of snake antivenom in Taiwan, including its incidence, risk factors, complications, and management. This could facilitate the development of preventive strategies and resource allocation. In Taiwan, six medically important venomous snake species were found; however, majority (73.3%) of the envenomation events were associated with *T. s. stejnegeri* (bamboo pit viper) and *P. mucrosquamatus* (Taiwan habu) and some (17.1%) with *N. atra* (Chinese cobra) and *B. m. multicinctus* (many-banded krait). The risks varied depending on the season, sex, age, geographic region, and occupation. Compared with patients envenomed by crotalines, those envenomed by elapids were more likely to undergo surgery due to serious wound complications caused by *N. atra* or endotracheal intubation due to respiratory failure caused by *B. m. multicinctus*. The low case-fatality rate observed might be due to easy access to modern medicine, immediate availability of antivenoms in healthcare facilities, few adverse effects of antivenoms, and prompt surgical intervention. The lack of highly venomous snakes with highly potent venom is a leading factor in low case fatality.

## Introduction

In 2017, the World Health Organization (WHO) reinstated snakebite into the list of "neglected tropical diseases" to emphasize its threats to humans [1]. More than 3,000 snake species has been discovered worldwide, around 600 of which are venomous [2]. The annual number of venomous snakebites is about 0.4–2.5 million [2,3]. The top three areas with the highest number of snakebites are South Asia, Sub-Saharan Africa, and Southeast Asia, with incidence rates ranging from 7.8 to 84.7 cases per 100,000 person-years [2]. The case-fatality rate of snakebite envenomation (SBE) due to serious complications or mismanagement is around 5% [2,3].

Snake species, human activities, health-seeking behavior, and appropriateness of the antivenom administration or surgical intervention are among the known factors associated with snakebite complications. Snakes inhabit area at different latitudes and altitudes, and their activities vary depending on environmental factors, such as season, temperature, and humidity

[4–8]. People bitten by snakes were more likely to be young men with more outdoor activities [5–15], especially those engaged in agricultural or pastoral labor [9,11,14–16]. According to the literatures, key factors associated with complications and even death from venomous snakebites include delay in seeking medical care [9,17], inappropriate pre-hospital care [9,15], incorrect snake species identification [18], insufficient antivenom administration [19,20], adverse effects of antivenom [12], and lack of designated treatment guidelines [21,22].

Taiwan is an island in southeastern Asia with an area of 33,883 km$^2$. The total population was 23,492,074, and the number of people in agriculture was 2,710,680 (11.5%) in 2022, respectively [23,24]. Six venomous snake species with medical importance were found: *Trimeresurus stejnegeri stejnegeri*, *Protobothrops mucrosquamatus*, *Deinagkistrodon acutus*, and *Daboia siamensis* in the Viperidae family and *Naja atra* and *Bungarus multicinctus multicinctus* in the Elapidae family [25]. The Taiwan Centers for Disease Control (CDC), the only snake antivenom manufacturer, produces four types of antivenom in lyophilized form, which are ammonium sulfate-precipitated F(ab')$_2$ fragments [26]. There are two bivalent specific antivenoms (one for *T. s. stejnegeri* and *P. mucrosquamatus*, and one for *N. atra* and *B. m. multicinctus*) and two monovalent antivenoms (one for *D. acutus* and one for *D. siamensis*). The diagnosis and management of SBEs followed the guidelines of the Taiwan Poison Control Center as no commercial venom serum test was available during the study period [26]. Briefly, the culprit snake was diagnosed by examining the snake brought or pictured by the patient or their associates. If not available, the diagnosis was made by asking the patient to identify the snake in a picture provided in the emergency department or by identifying typical manifestations through physical examination, serial wound inspection, and relevant medical history [27–33]. The recommended dosage of relevant antivenom to treat a envenomed patient is 1–2 vials for *T. s. stejnegeri*, 2–4 vials for *P. mucrosquamatus*, 2–4 vials for *D. acutus*, 2–4 vials for *D. siamensis*, 6–10 vials for *N. atra*, and 2–4 vials for *B. m. multicinctus* bite envenomings. The incidence rate of SBE decreased from 8.8 cases per 100,000 person-years during 1904–1938 to 4.5 cases per 100,000 person-years during 2005–2009; the case-fatality rate also decreased from 3%–7% to 0.04%–0.08% during the same period, respectively [34,35]. The decline in incidence was related to lifestyle changes, while the large reduction in case-fatality rate might be due to easy access to modern medicine, immediate availability of antivenom in healthcare facilities, and few adverse effects of antivenoms [25,26].

In Taiwan, nationwide and long-term epidemiological research on SBE remains lacking after 1940s [25]. This study aimed to explore the epidemiology of SBE based on the distribution and use of different antivenoms in different parts of Taiwan. The National Health Insurance Research Database (NHIRD) was used to investigate the demographics and clinical features of SBE to facilitate the development of prevention strategies and resource allocation.

## Methods

### Patient and public involvement

Patients or the public were not involved in the design, or conduct, or reporting, or dissemination plans of our research.

### Study population and setting

The NHI system provides uniform insurance coverage for everyone. This retrospective study was conducted using the NHIRD from 2002 to 2014. During this period, the Taiwan CDC distributed four types of antivenoms against the six medically important snakes to medical facilities: bivalent antivenom for *T. s. stejnegeri* and *P. mucrosquamatus* (FH), bivalent antivenom for *N. atra* and *B. m. multicinctus* (FN), and monovalent antivenom for *D. acutus* (FA);

monovalent antivenom for *D. siamensis* (FD) was not distributed until 2008 [25]. Patients could only obtain antivenom from the CDC through the NHI system, which is mandatory, and their medical records were detailed in the NHIRD. Those treated with the aforementioned antivenoms during 2002–2014 were included for analysis.

## Data collection

In the NHIRD, each drug or procedure was designated a specific code. We retrieved records from the NHIRD regarding the use of FH, FN, FA, and FD. Patient demographics, complications, management, and outcomes associated with the use of such antivenoms were then obtained. The diagnosis was adherent to the International Classification of Diseases (ICD)-9-Clinical Modification or ICD-10.

## Definition of the variables

Patient demographics included sex, age, comorbidity, occurrence date, geographic region of bite event, and occupation. The patients were divided into three groups: underage (<18 years), adults (18–65 years), and elderly (>65 years). The severity of the patient's comorbidity was stratified according to the Charlson Comorbidity Index (CCI) score [36]. Occurrence date was defined as the date of receipt of the first dose of antivenom. The months were categorized into spring (March, April, May), summer (June, July, August), autumn (September, October, November), and winter (December, January, February). Geographic region of bite event was defined as the location of the medical facility where the first dose of antivenom was administered, categorized into northern (Taipei, New Taipei, Keelung, Taoyuan, Hsinchu, Miaoli, Lianjiang, Kinmen), central (Taichung, Changhua, Nantou), southern (Yunlin, Chiayi, Tainan, Kaohsiung, Pingtung, Penghu), and eastern (Yilan, Hualien, Taitung) regions [37–39]. The forest area coverage of the region was defined as the ratio of the forest area to the land area based on aerial survey and field investigation [40]. Occupation was categorized into agricultural worker, laborer, and others.

A single envenomation event was arbitrarily defined as all records within 90 days after the first dose of antivenom. Anaphylactic reaction to antivenom administration was defined as being given epinephrine intravenously for shock, bronchospasm, or angioedema within 1 day following the first dose [41]. Wound infection and antibiotic use were defined as having erysipelas, cellulitis, fasciitis, or myositis and receiving oral or intravenous antibiotics within 7 days, respectively [42]. Surgery was defined as undergoing incision, division, excision, debridement, transplantation, reconstruction, grafts, or amputation within 14 days. Neurological complication was defined as experiencing paralysis, stroke, and encephalopathy within 30 days. Endotracheal intubation was defined as receiving invasive mechanical ventilation within 7 days. Death without other apparent causes within 90 days or during the same visit was defined as SBE related.

## Statistical analysis

Cumulative incidence was calculated annually as the total number of occurrences divided by the total number of individuals in the at-risk population. The directly standardized incidence was calculated based on the sex and age distribution in the demographic data from the Taiwan government and the 2000 World Standard Population from the WHO. Due to the skewed count data distributions, the Poisson regression model was used to analyse the relative risks (RRs) of the directly standardized incidence stratified by sex, age, season, geographic region, and occupation. Trend analysis in the general linear model was conducted to observe the annual changes. Spearman's rank correlation coefficient was calculated to check the agreement

on the ranking of the results between the cumulative incidence and forest area coverage in each county. A logistic regression model was used to identify factors related to the antivenom types and dosage, surgery, and death. Variables with a *P* value ≤0.25 in the univariable analysis were added in a stepwise manner; only those with a two-sided *P* value ≤0.05 were considered statistically significant and used in the final multivariable analysis. Sex, age, and CCI score were included in the final model regardless of the significance level. Statistical Analysis Software version 9.4 (SAS Institute, Cary, NC, USA) was used for the statistical analyses.

## Results

### Cumulative incidence

During 2002–2014, 12,542 patients were treated with antivenoms. The average number of cases per year was 965 (range, 871–1,072), and the crude incidence was 4.2 cases per 100,000 individuals (Table 1). The directly standardized incidence was 3.6 cases per 100,000 individuals, with a significant decrease from 4.5 cases per 100,000 individuals in 2002 to 3.1 cases per 100,000 individuals in 2014 ($p = 0.0002$) (Fig 1).

### Sex and age

Of the 12,542 patients, 70.1% were male, and 28.3% were female. The directly standardized incidences for men and women were 5.1 and 2.0 cases per 100,000 individuals, respectively, and the RR of men versus women was 2.5 ($p < 0.0001$) (Table 1). Their median age was 53

**Table 1. Characteristic distributions of SBE cases based on use of snake antivenom, and calculation of incidence and relative risk by characteristics (N = 12,542).**

| Variable | | Number | Percentage | Crude incidence (95% CI) | Directly standardized incidence (95% CI) | Relative risk (95% CI) | *p* value [2] |
|---|---|---|---|---|---|---|---|
| All cases | | 12,542 | 100.0% | 4.19 (4.12–4.27) | 3.59 (2.76–4.43) | N/A | N/A |
| Gender | Male | 8,791 | 70.1% | 5.83 (5.70–5.95) | 5.07 (3.67–6.48) | 2.51 (2.41–2.61) | <0.0001 |
| | Female | 3,544 | 28.3% | 2.39 (2.31–2.47) | 2.00 (1.12–2.88) | reference | |
| | Unknown | 207 | 1.7% | N/A | N/A | N/A | N/A |
| Age | <18 years | 459 | 3.7% | 0.73 (0.66–0.79) | N/A | reference | |
| | 18–64 years | 8,958 | 71.4% | 4.37 (4.28–4.46) | N/A | 5.99 (5.45–6.58) | <0.0001 |
| | ≥65 years | 3,125 | 24.9% | 10.05 (9.70–10.40) | N/A | 14.28 (12.94–15.75) | <0.0001 |
| Region | Northern | 4,326 | 34.5% | 3.24 (3.14–3.33) | 2.83 (1.70–3.95) | reference | |
| | Central | 1,863 | 14.9% | 3.22 (3.07–3.36) | 2.83 (1.13–4.53) | 1.02 (0.96–1.07) | 0.5350 |
| | Southern | 3,218 | 25.7% | 3.50 (3.38–3.63) | 2.87 (1.55–4.19) | 1.03 (0.99–1.08) | 0.1596 |
| | Eastern | 3,135 | 25.0% | 23.27 (22.45–24.08) | 19.56 (10.42–28.7) | 6.76 (6.45–7.08) | <0.0001 |
| Occupation | Agricultural worker | 5,045 | 40.2% | 16.03 (15.59–16.47) | 10.81 (5.53–16.09) | 5.53 (5.27–5.81) | <0.0001 |
| | Laborer | 3,108 | 24.8% | 1.96 (1.90–2.03) | 1.94 (1.00–2.87) | reference | |
| | Others | 4,389 | 35.0% | 3.99 (3.87–4.11) | 3.39 (2.02–4.76) | 1.77 (1.69–1.86) | <0.0001 |
| Types of antivenom [1] | FH | 9,194 | 73.3% | 3.07 (3.01–3.14) | 2.62 (1.91–3.33) | 38.43 (33.69–43.84) | <0.0001 |
| | FN | 2,150 | 17.1% | 0.72 (0.69–0.75) | 0.62 (0.27–0.97) | 8.97 (7.82–10.28) | <0.0001 |
| | FH + FN | 967 | 7.7% | 0.32 (0.30–0.34) | 0.29 (0.05–0.53) | 4.00 (3.46–4.63) | <0.0001 |
| | Others | 231 | 1.8% | 0.08 (0.07–0.09) | 0.07 (−0.05–0.18) | reference | |

[1] FH, bivalent antivenom against *T. s. stejnegeri* and *P. mucrosquamatus*; FN, bivalent antivenom against *N. atra* and *B. m. multicinctus*; Others, including antivenom against *D. acutus*, antivenom against *D. siamensis*, and their combination in administration.

[2] *p* value: The relative risk of directly standardized incidence using the Poisson regression model.

Abbreviations: CI, confidence interval; N/A, not applicable.

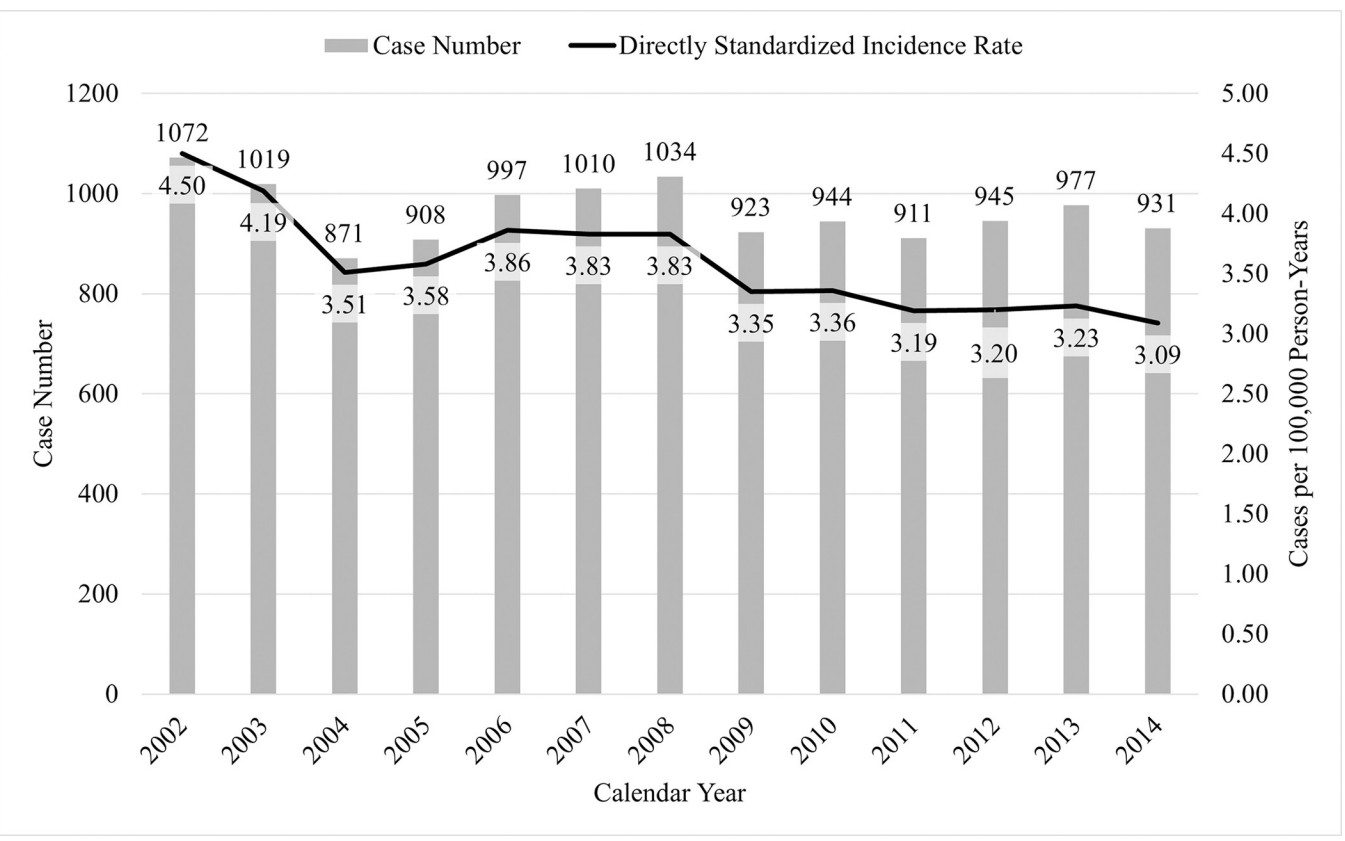

**Fig 1. The annual case number and directly standardized incidence of patients receiving antivenoms in Taiwan (*N* = 12,542).** *Data is available in S1 Data file.

years (men, 51 years; women, 57 years). The RRs of the elderly and adults versus the underage were 14.3 (*p* < 0.0001) and 6.0 (*p* < 0.0001), respectively (Table 1). Age exhibited an increasing trend, with an average increase of 0.5 years per year (*p* < 0.0001). The proportion of the elderly had gradually increased from 21.4% in 2002 to 27.6% in 2014.

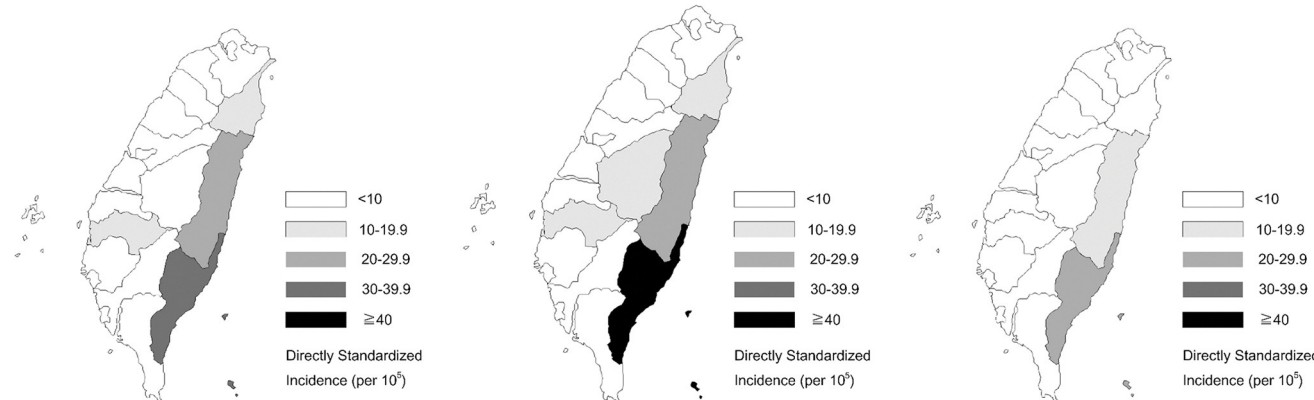

**Fig 2. The directly standardized incidence by region in patients receiving antivenoms in Taiwan (*N* = 12,542).** * Fig 2 was generated using Statistical Analysis Software version 9.4 (SAS Institute, Cary, NC, USA). The link to the base layer of the map is: https://data.gov.tw/dataset/7442. Data is available in S1 Data file.

**Table 2. Logistic regression analyses of the factors associated with FN antivenom administration with respect to FH antivenom administration (N = 12,542).**

| Variable | | FN (n = 2,150; 19.0%) | | FH (n = 9,194; 81.0%) | | $p_a$ value [1] | Univariable analysis | | Multivariable analysis | |
|---|---|---|---|---|---|---|---|---|---|---|
| | | n | % | n | % | | Crude odds ratio (95% CI) | $p_b$ value [1] | Adjusted odds ratio (95% CI) | $p_c$ value [1] |
| Male gender | | 1,636 | 76.1% | 6,365 | 69.2% | <0.0001 | 1.41 (1.27–1.58) | <0.0001 | 1.45 (1.29–1.62) | <0.0001 |
| Age | <18 years | 71 | 3.3% | 327 | 3.6% | <0.0001 | reference | | reference | |
| | 18–64 years | 1,656 | 77.0% | 6,425 | 69.9% | | 1.19 (0.91–1.54) | 0.1999 | 1.21 (0.91–1.61) | 0.1800 |
| | ≥65 years | 423 | 19.7% | 2,442 | 26.6% | | 0.80 (0.61–1.05) | 0.1098 | 0.85 (0.62–1.15) | 0.2926 |
| Charlson Comorbidity Index | 0 | 534 | 24.8% | 2,334 | 25.4% | 0.3235 | reference | | reference | |
| | 1 | 446 | 20.7% | 1,883 | 20.5% | | 1.04 (0.90–1.19) | 0.6268 | 1.09 (0.94–1.26) | 0.2477 |
| | 2 | 321 | 14.9% | 1,497 | 16.3% | | 0.94 (0.80–1.09) | 0.4059 | 1.00 (0.85–1.18) | 0.9787 |
| | ≥3 | 849 | 39.5% | 3,480 | 37.9% | | 1.07 (0.95–1.20) | 0.2954 | 1.26 (1.10–1.44) | 0.0010 |
| Season | Spring | 460 | 21.4% | 2,225 | 24.2% | <0.0001 | reference | | reference | |
| | Summer | 987 | 45.9% | 3,002 | 32.7% | | 1.59 (1.41–1.80) | <0.0001 | 1.46 (1.29–1.67) | <0.0001 |
| | Fall | 599 | 27.9% | 2,982 | 32.4% | | 0.97 (0.85–1.11) | 0.672 | 0.91 (0.79–1.05) | 0.2077 |
| | Winter | 104 | 4.8% | 985 | 10.7% | | 0.51 (0.41–0.64) | <0.0001 | 0.46 (0.36–0.58) | <0.0001 |
| Region | Northern | 483 | 22.5% | 3,299 | 35.9% | <0.0001 | reference | | reference | |
| | Central | 462 | 21.5% | 1,235 | 13.4% | | 2.56 (2.21–2.95) | <0.0001 | 2.59 (2.22–3.02) | <0.0001 |
| | Southern | 846 | 39.3% | 2,130 | 23.2% | | 2.71 (2.40–3.07) | <0.0001 | 3.21 (2.81–3.67) | <0.0001 |
| | Eastern | 359 | 16.7% | 2,530 | 27.5% | | 0.97 (0.84–1.12) | 0.6746 | 1.02 (0.87–1.18) | 0.8289 |
| Occupation | Agricultural worker | 795 | 37.0% | 3,856 | 41.9% | <0.0001 | 0.78 (0.69–0.88) | <0.0001 | 0.61 (0.53–0.71) | <0.0001 |
| | Laborer | 579 | 26.9% | 2184 | 23.8% | | reference | | reference | |
| | Others | 776 | 36.1% | 3154 | 34.3% | | 0.93 (0.82–1.05) | 0.2252 | 0.89 (0.78–1.01) | 0.0824 |

[1] $p_a$ value: the χ2 test was used to compare categorical variables; $p_b$ value: Univariable analysis using the logistic regression model; $p_c$ value: Multivariable analysis using the logistic regression model.

Abbreviations: CI, confidence interval; FH, bivalent antivenom against *T. s. stejnegeri* and *P. mucrosquamatus*; FN, bivalent antivenom against *N. atra* and *B. m. multicinctus*.

## Month of occurrence

The incidence peaked in August, which was summer, accounting for 13.4% of all envenomation events. Contrarily, January had the lowest incidence, which was winter, accounting for 2.5% of all envenomation events. The incidences of patients given FH were evenly distributed during May–November, whereas those of patients given FN, FA, and FD were during July–September, August, and June–July, respectively.

## Geographic region

The proportions of cases in northern, central, southern, and eastern Taiwan were 34.5%, 14.9%, 25.7%, and 25.0%, respectively. The directly standardized incidence in eastern Taiwan was the highest, which was 19.6 per 100,000 individuals, whereas those in northern, central, and southern Taiwan were 2.8, 2.8, and 2.9 cases per 100,000 individuals, respectively (Fig 2). The RR of envenomation in eastern Taiwan versus northern Taiwan was 6.8 ($p < 0.0001$) (Table 1). The geographic variation in incidence was found to be possibly related to the forest area coverage, with a Spearman's rank correlation of 0.65 between cumulative incidence and forest area coverage in each county ($p = 0.0025$).

## Occupation

Of the 12,542 patients, 40.2% were agricultural workers and 24.8% were laborers. The directly standardized incidences for agricultural workers and laborers were 10.9 and 1.9 cases per 100,000 individuals, respectively, and the RR of agricultural workers versus laborers was 5.5 ($p < 0.0001$) (Table 1). Contrary to the annual decrease of 0.06 cases per 100,000 individuals among laborers ($p = 0.0018$), the annual incidence among agricultural workers did not significantly change during the study period ($p = 0.3524$).

Agricultural workers and laborers were found to have different characteristics. Among agricultural workers, the male-to-female ratio was 1.5, whereas among laborers, it was 2.3. Furthermore, the median age of agricultural workers was 60 years, whereas that of laborers was 44 years. Also, the proportions of agricultural workers in northern, central, southern, and eastern Taiwan were 22.6%, 19.7%, 35.9%, and 21.8%, respectively, whereas for laborers, the proportions were 50.9%, 12.0%, 17.8%, and 19.3%, respectively.

## Type and dose of antivenom

The proportions of patients given FH, FN, FA, FD, and mixed antivenoms were 73.3%, 17.1%, 0.6%, 0.1%, and 8.8%, respectively (S1 Table); 98.2% of the patients received at least one dose of FH or FN. Compared with patients given FH, those given FN were more likely to be male, bitten during the summer, in central or southern Taiwan, and laborer (Table 2). The average doses of patients given FH, FN, FA, FD, and mixed antivenoms were 2.5, 3.4, 3.0, 2.4, and 5.6 vials, respectively. Furthermore, the proportions of patients given 1, 2, 3–4, and ≥5 vials of antivenom were 42.8%, 22.9%, 17.5%, and 16.9%, respectively. The dose of antivenom in a patient ranged from 1 to 40 vials, with an average of 3.0 vials; moreover, a significant increase of 0.04 vials per person per year was observed ($p = 0.0189$).

## Complications and management

Of the 12,542 patients, only 0.1% experienced anaphylactic shock and was treated with epinephrine. Although systemic antibiotics were administered to as high as 81.8% of the patients, only 21.7% received antibiotics for a defined infection, and only 1.7% were diagnosed with necrotizing fasciitis. Among those given antibiotics, 70.0% received cephalosporins or penicillins alone, and 21.0% received cephalosporins or penicillins plus aminoglycosides. Surgery was performed in 7.3% of the patients, including incision or division in 2.7%, excision or debridement in 6.1%, transplantation or reconstruction in 2.8%, and amputation in 0.3%. Neurological complications occurred in 1.4%, and 0.7% underwent endotracheal intubation.

In the multivariable analyses, the antivenom type was not associated with anaphylactic shock; however, patients given FN were more likely to have infections (adjusted odds ratio [aOR] = 1.19, $p = 0.0036$), necrotizing fasciitis (aOR = 7.80, $p < 0.0001$), surgery (aOR = 3.36, $p < 0.0001$), neurological complications (aOR = 3.10, $p < 0.0001$), and endotracheal intubation (aOR = 9.26, $p < 0.0001$) than those given FH.

## Death

A total of 14 patients died, with a case-fatality rate of 0.11%. Among them, eight and six patients died after receiving FH and FN, with case-fatality rates of 0.09% and 0.28%, respectively. Of the 14 patients, 6 died within 7 days, and 8 died within 52 days. The main causes of death were renal failure, intracranial hemorrhage, and respiratory failure. No one died from anaphylactic shock or necrotizing fasciitis.

## Discussion

Under-reporting of snakebite incidence and mortality is common in South and Southeast Asian countries and many SBEs land up with alternate therapies. In Nepal, for example, hospital reports suggested that about 20,000 patients were admitted and 1,000 died from snakebites each year, yet figures collected in a community based study suggested that 26,749–37,661 people bitten by snakes and 2,386–3,225 deaths per year across only the Terai region [43]. Similarly, hospital records revealed approximately one third of snakebite incidents found in the community survey in Vietnam [44]. We therefore utilized NHIRD based on antivenom distribution and use to analyze patients envenomed by snakes though the actual epidemiology of SBE would be somewhat different because there is a possibility that some mildly envenomed patients may not receive antivenom, or some patients may have received antivenom inappropriately. However, alternative therapies for SBE do not occur or are extremely rare in Taiwan. Traditional healers are illegal because all medical practices are regulated by laws. Therefore, we believe our study findings are representative for individuals with SBE.

A decrease in the directly standardized incidence of patients given antivenoms was observed during the study period. Patients envenomed by *T. s. stejnegeri* and *P. mucrosquamatus* (i.e., treated with FH) were predominant, followed by patients envenomed by *N. atra* and *B. m. multicinctus* (i.e., treated with FN). The factors found to be associated with the increased risk of SBE were male gender, old age, summer season, being in eastern Taiwan, and being an agricultural worker. In patients envenomed by *N. atra* and *B. m. multicinctus*, the risk factors were male gender, summer season, being in central or southern Taiwan, and being a laborer. Compared with patients envenomed by *T. s. stejnegeri* and *P. mucrosquamatus*, those envenomed by *N. atra* and *B. m. multicinctus* were more likely to have infections, necrotizing fasciitis, surgery, neurological complications, and endotracheal intubation. However, the case-fatality rate for any venomous snakebite was quite low.

This study demonstrated that the directly standardized incidence in Taiwan was 3.6 cases per 100,000 persons, which was relatively low among Asian countries [2]. There were 2.5 times as many male patients as female patients, similar to Taiwan in the 1940s [34] and other countries in Southeast Asia [10–13], but slightly higher than in East and South Asia [5–9], presumably representing the ratio of males to females engaging in outdoor work or activities. The patients' median age was 53 years, similar to that in East Asia [6–8] but significantly higher than that in Taiwan in the 1940s [34], Southeast Asia [10–13], and South Asia [5,9]. The increasing age of the patients was consistent with the aging population in Taiwan.

SBE occurred more frequently during the warm seasons [45], consistent with previous studies [5–8,34,35,37,46–52]. *T. s. stejnegeri* and *P. mucrosquamatus* envenomations were frequent during June–November, whereas *N. atra* and *B. m. multicinctus* envenomations mainly peaked during June–August. SBE was at least five times more likely to occur among agricultural workers than laborers; snakebite could also be a significant occupational injury associated with agricultural activities [9,11,14,16]. With the industrial transformation in Taiwan since the 1950s, young people were more inclined to engage in non-agricultural work, whereas the elderly who remained as agricultural workers seemed to be at the highest risk of SBE. In eastern Taiwan, which has large forest area coverage, the incidence of SBE was at least six times that in other areas, due to not only increased exposure to snakes because of agricultural work but also the development of sightseeing agriculture and popularity of recreational tourism in forest areas.

In this study, FH and FN were used in majority of the patients, consistent with the results of previous studies in Taiwan; contrarily, patients treated with FA or FD were much fewer than previously reported [34,37]. *T. s. stejnegeri* and *P. mucrosquamatus* envenomations were predominant throughout the island, and *N. atra* and *B. m. multicinctus* were identified as critical

species in central and southern Taiwan [34,37,47]. Although *N. atra* appeared more frequently in lower-altitude farmlands than *T. s. stejnegeri* and *P. mucrosquamatus* [25,34,37,47], agricultural workers were more likely to receive FH than laborers. This might be because agricultural workers who had been working in the farmland for a long time were more familiar with the habits of *N. atra* and were not easily bitten.

In Taiwan, the main effects of *T. s. stejnegeri* and *P. mucrosquamatus* bites were tissue swelling, pain, and local ecchymosis, whereas systemic effects were uncommon [32,33]. *N. atra* bites could cause local tissue necrosis and secondary wound infection, whereas *B. m. multicinctus* bites could cause neuromuscular paralysis leading to respiratory failure [27,30]. Although snake species could not be specified based on the use of bivalent antivenom, our study findings indicated that patients given FN were more likely to undergo surgery due to serious wound complications caused by the cytotoxins in *N. atra* venom [30,31,53,54] or endotracheal intubation due to respiratory failure caused by *B. m. multicinctus* envenomation [27].

Patients may require surgery due to local tissue necrosis, wound infections such as necrotizing fasciitis, or inappropriate first-aid measures [9,55]. Overall, 7.3% of the patients in this study underwent surgery, which was lower than those in other Asian [7,53,56–58] and sub-Saharan African [15,59,60] countries but similar to those in European and American countries [61–64], although guidelines on the timing of surgical intervention after snakebites are inconclusive [31,41,65–68]. To reduce the need for surgery, patients should receive an adequate amount of antivenom [69,70].

Diagnosing wound infections in SBEs can be difficult and may be overestimated if deep tissue or biopsy cultures are not performed for the wound. However, we observed that 76.8% to 80.9% of *N. atra* bite patients developed wound infections in studies conducted by Mao et al [30,71]. Out of the 59 and 87 patients, 50 and 73, respectively, had positive wound bacterial cultures [30,71]. In addition, 22 out of 27 and 30 out of 33 had positive deep tissue or biopsy cultures, respectively [30,31]. Furthermore, necrotizing soft tissue infections, including necrotizing fasciitis, were observed in 42.1% to 44.7% of *N. atra* bites [30,31]. On the other hand, wound infections are less common in *T. s. stejnegeri* and *P. mucrosquamatus* (crotaline) bites, with rates of 11.4% and 27.4%, respectively [32,33]. In this study, which utilized the NHIRD, we used criteria similar to those in a recent report to identify SBE wound infections [42]. Our findings indicate that 20.3% of patients who received FH antivenom (i.e., antivenom for *T. s. stejnegeri* and *P. mucrosquamatus*) developed wound infections, which is consistent with previous studies conducted by Mao et al. [32,33], and lower than the study conducted by João Victor Soares Coriolano Coutinho et al.[42]. Although the incidence of wound infection might still have been overestimated in this study, we believed the overestimate was likely to be of limited magnitude.

During 2002–2014, the case-fatality rate of SBE in Taiwan was 0.11%, which was lower than that in Taiwan in the early 20[th] century (3%–7%) [34,37,48–50] or in other countries (0.2%–58%) [5,9–11,17,72–79]. In this study, patients envenomed by *N. atra* or *B. m. multicinctus* had a higher case-fatality rate than those envenomed by *T. s. stejnegeri* or *P. mucrosquamatus*, as in previous studies [34,37,48–50]. In general, the high number of deaths from SBE was due to delayed or inappropriate antivenom treatment [9,10,17,72,74], and more than half of the deaths were from shock due to bleeding, capillary leak syndrome, or myocardial depression [10,73,74]. In Taiwan, most of the damage caused by SBE was not fatal due to the availability of antivenom and modern medicine; nor did anyone die from complications that could be aggressively managed, such as anaphylactic shock and necrotizing fasciitis.

## Limitations

This study has several limitations. First, the drug code of snake antivenom was used to obtain information from the NHIRD, excluding patients who died without seeking medical care, with

mild symptoms not requiring antivenom therapy, and envenomed by *D. siamensis* before 2008. Seven other venomous snakes were found in Taiwan: *Ovophis makazayazaya* and *T. gracilis* in the Viperidae family, *Sinomicrurus swinhoei* and *S. sauteri* in the Elapidae family, *Rhabdophis tigrinus formosanus* and *Pseudagkistrodon rudis* in the Colubridae family, and *Myrrophis chinensis* in the Homalopsidae family [26,34,80–87]. However, the rarity of bites by these snakes makes it unlikely for them to significantly affect our findings. Second, the individual SBE (i.e., *T. s. stejnegeri*, *P. mucrosquamatus*, *N. atra*, and *B. m. multicinctus*) could not be determined as the analysis was based on the use of bivalent antivenom against these species. In addition, the decision to use antivenom, or a different type of antivenom, was made by clinicians based on the snake identification procedure, without confirmation of venom in the serum, which could introduce bias. It is also possible that some victims received antivenom for a snakebite, which was later found to be a non-venomous bite or different type of bite. These determinations can only be confirmed through a thorough review of individual case notes. Third, we defined all antivenom records within 90 days as one event and thus might have missed a few cases with repeated SBE. The case number in our study was 5% more than those of the study without definition [35] and 30% less than those of the study defining antivenom records within 14 days as one event [47]. Fourth, the geographic regions of snakebite incidents may be biased. Due to the small land area and convenient transportation in Taiwan, the location of the hospital where patients sought help did not necessarily represent the location where they were envenomed. Furthermore, the patients' insurance status might not accurately represent their current occupation. Factors affecting the epidemiology of snakebites, such as geographic region and human activities, need to be further studied. Finally, because this study is retrospective, which has certain inherent limitations, the results should be interpreted cautiously.

## Conclusions

This study described the epidemiologic and local and overall systemic toxin effects of SBE in Taiwan based on data using antivenom. Risk factors such as male gender, old age, summer season, being in eastern Taiwan, and being an agricultural worker. It is worth noting that elapid bites, which caused necrotizing fasciitis and respiratory failure more than crotaline bites, were more likely to occur in central or southern Taiwan and among laborers. Differences of the epidemiological findings between snake species should be focused on when developing strategies for snakebite prevention.

## Supporting information

**S1 Table. Characteristics of patients who received different antivenoms in Taiwan (*N* = 12,542).**
(DOCX)

**S1 Data. Fig 1.** The annual case number and directly standardized incidence of patients receiving antivenoms in Taiwan (*N* = 12,542). Fig 2. The directly standardized incidence by region in patients receiving antivenoms in Taiwan (*N* = 12,542).
(XLSX)

## Acknowledgments

The authors thank Jou-Fang Deng, the founder of Taiwan Poison Control Center, for his inspiration of this study.

## Author Contributions

**Conceptualization:** Jen-Yu Hsu, Yan-Chiao Mao.

**Data curation:** Jen-Yu Hsu, Shu-O Chiang, Yan-Chiao Mao.

**Formal analysis:** Jen-Yu Hsu, Shu-O Chiang, Tan-Wen Hsieh.

**Funding acquisition:** Yan-Chiao Mao.

**Investigation:** Jen-Yu Hsu, Shu-O Chiang, Chen-Chang Yang, Tan-Wen Hsieh, Chi-Jung Chung, Yan-Chiao Mao.

**Methodology:** Jen-Yu Hsu, Shu-O Chiang, Chen-Chang Yang, Tan-Wen Hsieh, Chi-Jung Chung, Yan-Chiao Mao.

**Project administration:** Yan-Chiao Mao.

**Resources:** Yan-Chiao Mao.

**Software:** Jen-Yu Hsu, Shu-O Chiang, Tan-Wen Hsieh.

**Supervision:** Chen-Chang Yang, Chi-Jung Chung, Yan-Chiao Mao.

**Validation:** Jen-Yu Hsu, Shu-O Chiang, Chen-Chang Yang, Tan-Wen Hsieh, Chi-Jung Chung, Yan-Chiao Mao.

**Visualization:** Jen-Yu Hsu, Shu-O Chiang, Yan-Chiao Mao.

**Writing – original draft:** Jen-Yu Hsu.

**Writing – review & editing:** Yan-Chiao Mao.

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
