## [Decision Letter · Decision Letter 0]

12 Apr 2023

Dear Dr. Mao,

Thank you very much for submitting your manuscript "Nationwide and long-term epidemiological research of snake envenomation in Taiwan during 2002–2014: a study utilizing National Health Insurance Database" for consideration at PLOS Neglected Tropical Diseases. As with all papers reviewed by the journal, your manuscript was reviewed by members of the editorial board and by several independent reviewers. In light of the reviews (below this email), we would like to invite the resubmission of a significantly-revised version that takes into account the reviewers' comments. 

The manuscript adds important information to the field. It will be great to add actual epidemiological data on snakebite envenomation rather than the surrogates used. 

Because of its importance I recommend that the authors revise the manuscript after major revision and resubmit.

We cannot make any decision about publication until we have seen the revised manuscript and your response to the reviewers' comments. Your revised manuscript is also likely to be sent to reviewers for further evaluation.

Sincerely,

R. Manjunatha Kini

Academic Editor

Justin Remais

Section Editor

The manuscript adds important information to the field. It will be great to add actual epidemiological data on snakebite envenomation rather than the surrogates used. 

Because of its importance I recommend that the authors revise the manuscript after major revision and resubmit.

Reviewer's Responses to Questions

**Key Review Criteria Required for Acceptance?**

**Methods**

-Are the objectives of the study clearly articulated with a clear testable hypothesis stated?

-Is the study design appropriate to address the stated objectives?

-Is the population clearly described and appropriate for the hypothesis being tested?

-Is the sample size sufficient to ensure adequate power to address the hypothesis being tested?

-Were correct statistical analysis used to support conclusions?

-Are there concerns about ethical or regulatory requirements being met?

Reviewer #1: Please refer the comments under Summary and General Comments

Reviewer #2: -the incidence of venomous snakebite being extrapolated from hospital insurance payouts maybe fallacious as many of the envenomed may actually land up with alternate therapies which fall below the insurance radar as so commonly happens in South-East Asian countries and whether these numbers reporting for treatment in modern medical facilities are thus truly representative enough

- it is not stated as to whether the e Taiwan National Health Insurance covers for all individuals and all diseases or in short whether the health system is uniformly insured for all

- There would be instances of victims being given ASV for a presumptive bite which would finally turn out as non-venomous or other bites which could only be confirmed with study of the individual case notes

**Results**

-Does the analysis presented match the analysis plan?

-Are the results clearly and completely presented?

-Are the figures (Tables, Images) of sufficient quality for clarity?

Reviewer #1: Please refer the comments under Summary and General Comments

Reviewer #2: The incidence of venomous snakebite may not be truly representative but a pointer towards the results given

**Conclusions**

-Are the conclusions supported by the data presented?

-Are the limitations of analysis clearly described?

-Do the authors discuss how these data can be helpful to advance our understanding of the topic under study?

-Is public health relevance addressed?

Reviewer #1: Please refer the comments under Summary and General Comments

Reviewer #2: Reasonably well, yes

**Editorial and Data Presentation Modifications?**

Reviewer #1: None

Reviewer #2: this manuscript on a Neglected Tropical disease is presented well given that it has all the negatives that come alongside retrospective data analysis using an Insurance database

I would think it could be accepted with revisions and of modest interest to readers

**Summary and General Comments**

Reviewer #1: Suggest to change the “snake envenomation” as “snakebite envenomation” in the title and throughout the manuscript. 

The terms “envenoming” and envenomation” have been used is different parts of the manuscript. Suggest to use single terminology throughout the manuscript.

Abstract: Reword the introduction of the abstract to highlight the lack of long-term epidemiological data of snakebite envenomation (SBE). It is not necessary to include the names of the venomous snakes with in the limited word count in the abstract.

This is not a true prospectively conducted SBE epidemiology study. All epidemiological results were generated based on the distribution and used of different antivenoms in different parts of Taiwan. Therefore, reword this sentence “This study aimed to explore the epidemiology of snake envenomation in Taiwan to facilitate the development of prevention strategies and resource allocation.” to highlight that this study is based on Antivenom data.

Add thousand separators in all the numerical values more than 999. (throughout the manuscript).

Authors summary: Include the common names of the listed snake species.

Authors summary: “The low case-fatality rate observed might be due to easy access to modern medicine, immediate availability of antivenoms in healthcare facilities, few adverse effects of antivenoms, and prompt surgical intervention.” 

Lack of highly venomous snakes have highly potent venom is a leading factor for low case fatality. Please include that in this sentence.

Under introduction: Following details on Taiwan and related to SBE have to be included.

Total land area, total population, proportion of people in rural areas, proportion of people in agricultural sector, type of snake antivenom used, antivenom manufacturers, details of the snakebite management guidelines used to treat patients, basis of deciding type of antivenom, how snakes are identified in hospitals. 

At the end of introduction: Aims and objectives of this study have not been stated.

Under study population and setting: include the details of antivenom manufacturers. 

Under study population and setting: “We assumed that patients who received specific snake antivenom were envenomed by relevant snakes”. Majority of the data analysis of this study depends on accurate identification of snake bitten by and correct used of antivenom. Therefore, it is important to provide more details about how bitten snakes were identified and antivenom selection procedure. 

The presented epidemiological data on SBE is based on antivenom usage database and which is far from the actual SBE epidemiology described by a prospective observation study. There is a possibility of some envenomated patients may not seek hospital admissions, some patients may not give the antivenom, some patients may have received antivenom inappropriately. Therefore, I would like to suggest to present this data as “use of snake antivenom in Taiwan”. This factor has not been highlighted because of using “epidemiology of snake envenomation” in several places. I suggest to reword the title and other relevant sections accordingly. 

Discussion: Highlight that all these epidemiological data is based on the Antivenom distribution and used data and the actual SBE epidemiology would be somewhat different and explain the reasons for the difference. 

Provide some SBE epidemiology figures in some Asian countries and discuss the Taiwan values with the values of some Asian countries. 

Limitations:

Describe the issues of accurate snake identification and decision of use of different type of Antivenom based on that.

“Second, the epidemiology of individual snakebites….” Here this is not on snakebites, snakebite envenomation based on antivenom using data. 

Conclusions: Second line “clinical features”

This study has not described the detailed clinical features of SBE. Presented overall local and systemic effects such as neurotoxicity, renal injury, and so on. Hence, this has to be modified as “local and overall systemic toxin effects”

Table 1. Caption: “Characteristic distributions of venomous snakebites, and calculation of…”

This is not snakebites, this has to be modified as “Characteristic distributions of SBE cases based on use of snake antivenom, and calculation of….

Reviewer #2: I read the manuscript written by Jen-Yu Hsu et al with a lot of interest and have the following remarks to make

-in the author summary he states that venomous snakebite is a tropical disease and should not be neglected, whilst venomous snakebite is already listed as a neglected tropical disease (NTD)

- Bungarus multicinctus multicinctus is I believe not distributed in India

- ‘Wound infection and antibiotic use were defined as having erysipelas, cellulitis, fasciitis, or myositis and receiving oral or intravenous antibiotics within 7 days, respectively’, would not hold true for all fasciitis or myositis as a lot of it could be attributed directly to the venoms of viperid, crotalid and even elapid species as well

- I wonder how many of the envenomed actually land up with alternate therapies which fall below the insurance radar as so commonly happens in South-East Asian countries and whether these numbers reporting for treatment in modern medical facilities are thus truly representative enough

- Given the constraints placed by retrospective analysis of data and that it being based on insurance pay-outs I think that the manuscript is structured well and makes for a good read.

PLOS authors have the option to publish the peer review history of their article (what does this mean?). If published, this will include your full peer review and any attached files.

Reviewer #1: Yes: Kalana Maduwage

Reviewer #2: Yes: Jaideep Menon
---

## [Editor Report · Decision Letter 1]

11 May 2023

Dear Dr. Mao,

Thank you very much for submitting your manuscript "Nationwide and long-term epidemiological research of snakebite envenomation in Taiwan during 2002–2014 based on the use of snake antivenoms: a study utilizing National Health Insurance Database" for consideration at PLOS Neglected Tropical Diseases. As with all papers reviewed by the journal, your manuscript was reviewed by members of the editorial board and by several independent reviewers. The reviewers appreciated the attention to an important topic. Based on the reviews, we are likely to accept this manuscript for publication, providing that you modify the manuscript according to the review recommendations. 

The revised manuscript satisfactorily responds to all the questions, queries and corrections suggested by the reviewers. Some parts of their rebuttal will help improve the manuscript. For example, segments in the rebuttal letter such as "We agree with you that diagnosing wound infections in SBEs can be difficult and may be overestimated if deep tissue or biopsy cultures are not performed for the wound. However, we observed that 76.8% to 80.9% of N. atra bite patients developed wound infections in studies conducted by Mao et al. Out of the 59 and 87 patients, 50 and 73, respectively, had positive wound bacterial cultures [1,2]. In addition, 22 out of 27 and 30 out of 33 had positive deep tissue or biopsy cultures, respectively [2,3]. Furthermore, necrotizing soft tissue infections, including necrotizing fasciitis, were observed in 42.1% to 44.7% of N. atra bites [2,3]. On the other hand, wound infections are less common in T. s. stejnegeri and P. mucrosquamatus (crotaline) bites, with rates of 11.4% and 27.4%, respectively [4,5]. In this study, which utilized the National Health Insurance database, we used criteria similar to those in a recent report to identify SBE wound infections [6]. Our findings indicate that 20.3% of patients who received FH antivenom (i.e., antivenom for T. s. stejnegeri and P. mucrosquamatus) developed wound infections, which is consistent with previous studies conducted by Mao et al. [4,5], and lower than the study conducted by João Victor Soares Coriolano Coutinho et al. [6]. Although the incidence of wound infection might still have been overestimated in this study, such overestimation is much smaller" along with the references. You could use such answers to reviewers' comments to improve the revised manuscript.

Sincerely,

R. Manjunatha Kini

Academic Editor

Justin Remais

Section Editor

The revised manuscript satisfactorily responds to all the questions, queries and corrections suggested by the reviewers. Some parts of their rebuttal will help improve the manuscript. For example, segments in the rebuttal letter such as "We agree with you that diagnosing wound infections in SBEs can be difficult and may be overestimated if deep tissue or biopsy cultures are not performed for the wound. However, we observed that 76.8% to 80.9% of N. atra bite patients developed wound infections in studies conducted by Mao et al. Out of the 59 and 87 patients, 50 and 73, respectively, had positive wound bacterial cultures [1,2]. In addition, 22 out of 27 and 30 out of 33 had positive deep tissue or biopsy cultures, respectively [2,3]. Furthermore, necrotizing soft tissue infections, including necrotizing fasciitis, were observed in 42.1% to 44.7% of N. atra bites [2,3]. On the other hand, wound infections are less common in T. s. stejnegeri and P. mucrosquamatus (crotaline) bites, with rates of 11.4% and 27.4%, respectively [4,5]. In this study, which utilized the National Health Insurance database, we used criteria similar to those in a recent report to identify SBE wound infections [6]. Our findings indicate that 20.3% of patients who received FH antivenom (i.e., antivenom for T. s. stejnegeri and P. mucrosquamatus) developed wound infections, which is consistent with previous studies conducted by Mao et al. [4,5], and lower than the study conducted by João Victor Soares Coriolano Coutinho et al. [6]. Although the incidence of wound infection might still have been overestimated in this study, such overestimation is much smaller" along with the references. You could use such answers to reviewers' comments to improve the revised manuscript.

Figure Files:

Data Requirements:

Reproducibility:

References

---

## [Editor Report · Decision Letter 2]

12 May 2023

Dear Dr. Mao,

We are pleased to inform you that your manuscript 'Nationwide and long-term epidemiological research of snakebite envenomation in Taiwan during 2002–2014 based on the use of snake antivenoms: a study utilizing National Health Insurance Database' has been provisionally accepted for publication in PLOS Neglected Tropical Diseases.

Best regards,

R. Manjunatha Kini

Academic Editor

Justin Remais

Section Editor

The authors have revised the manuscript to include all suggestions/clarifications.

---

## [Editor Report · Acceptance letter]

24 May 2023

Dear Dr. Mao,

We are delighted to inform you that your manuscript, "Nationwide and long-term epidemiological research of snakebite envenomation in Taiwan during 2002–2014 based on the use of snake antivenoms: a study utilizing National Health Insurance Database," has been formally accepted for publication in PLOS Neglected Tropical Diseases.

Best regards,

Shaden Kamhawi

co-Editor-in-Chief

Paul Brindley

co-Editor-in-Chief
